# Chemerin in Participants with or without Insulin Resistance and Diabetes

**DOI:** 10.3390/biomedicines12040924

**Published:** 2024-04-22

**Authors:** Lei Zhao, Jonathan Zhou, Fahim Abbasi, Mohsen Fathzadeh, Joshua W. Knowles, Lawrence L. K. Leung, John Morser

**Affiliations:** 1Division of Hematology, Stanford University School of Medicine, Stanford, CA 94305, USA; lawrence.leung@stanford.edu; 2Veterans Affairs Palo Alto Health Care System, Palo Alto, CA 94304, USA; 3University Program in Genetics and Genomics, School of Medicine, Duke University, Durham, NC 27705, USA; jonathan.zhou@duke.edu; 4Division of Cardiovascular Medicine, Stanford University School of Medicine, Stanford, CA 94305, USA; fahim@stanford.edu (F.A.); mohsen@stanford.edu (M.F.); knowlej@stanford.edu (J.W.K.)

**Keywords:** chemerin, diabetes, insulin, plasma glucose level

## Abstract

Chemerin is a chemokine/adipokine, regulating inflammation, adipogenesis and energy metabolism whose activity depends on successive proteolytic cleavages at its C-terminus. Chemerin levels and processing are correlated with insulin resistance. We hypothesized that chemerin processing would be higher in individuals with type 2 diabetes (T2D) and in those who are insulin resistant (IR). This hypothesis was tested by characterizing different chemerin forms by specific ELISA in the plasma of 18 participants with T2D and 116 without T2D who also had their insulin resistance measured by steady-state plasma glucose (SSPG) concentration during an insulin suppression test. This approach enabled us to analyze the association of chemerin levels with a direct measure of insulin resistance (SSPG concentration). Participants were divided into groups based on their degree of insulin resistance using SSPG concentration tertiles: insulin sensitive (IS, SSPG ≤ 91 mg/dL), intermediate IR (IM, SSPG 92–199 mg/dL), and IR (SSPG ≥ 200 mg/dL). Levels of different chemerin forms were highest in patients with T2D, second highest in individuals without T2D who were IR, and lowest in persons without T2D who were IM or IS. In the whole group, chemerin levels positively correlated with both degree of insulin resistance (SSPG concentration) and adiposity (BMI). Participants with T2D and those without T2D who were IR had the most proteolytic processing of chemerin, resulting in higher levels of both cleaved and degraded chemerin. This suggests that increased inflammation in individuals who have T2D or are IR causes more chemerin processing.

## 1. Introduction

Chemerin, encoded by the retinoic acid receptor responder 2 (*RARRES2*) gene, was identified in human inflammatory fluids as a natural ligand for the orphan G protein-coupled chemokine-like receptor 1 (CMKLR1), also known as chemR23 and now named chem1 [1,2]. It functions as a chemoattractant for leukocytes expressing chem1, such as plasmacytoid dendritic cells and natural killer cells. Two additional receptors bind chemerin with high affinity, chemokine receptor-like 2 (CCRL2) [3,4], and G protein-coupled receptor 1 (GPR1; chem2) [5]. CCRL2 is not a signaling receptor and only binds chemerin, presenting it to its other receptors [6]. In addition to its immune functions, chemerin is also an adipokine that regulates adipocyte development and metabolic functions such as glucose metabolism [7,8,9,10,11] with both chem1 and chem2 implicated in that role [12,13]. Elevated levels of chemerin have been found in patients with diabetes [14,15] and fatty liver disease [16,17,18]. Although chemerin levels in the blood are elevated in obese humans and rodents and chemerin may serve as a chemoattractant for various types of immune cells that contribute to adipose tissue inflammation commonly found with obesity, the relationship between obesity, insulin resistance, inflammation, and energy homeostasis in determining chemerin levels has not been defined [19].

A hallmark of cardiometabolic syndrome is insulin resistance, in which chemerin has been implicated in mouse experiments by observation of deterioration of insulin tolerance and glucose tolerance in chemerin or chem2 deficient mice [10,12,20,21]. Chemerin stimulates insulin-dependent glucose uptake concomitant with the enhanced insulin signaling in adipocytes [11]. In addition, chemerin drives differentiation of both white and brown adipose tissue in vitro and in vivo [8,20].

Chemerin is secreted from cells as a 143 amino acid protein with low activity named chem163S (different forms of chemerin are named for their C-terminal amino acid and residue number), that is subsequently enzymatically processed by serine proteases at its C-terminus to generate a partially active form, chem158K [22]. Then the C-terminal amino acid is removed by plasma basic carboxypeptidases to produce the fully active forms, chem157S and chem156F, which can also be made directly by chymase cleavage of chem163S [23,24,25]. Further proteolysis leads to chem155A and smaller forms such as chem144D that are inactive [26]. The enzymes responsible for processing chemerin are members of the coagulation, fibrinolytic, and inflammatory systems. Chemerin’s C terminal sequence and its proteolytic cleavage sites are highly conserved between human and mouse, as well as in other mammalian species, with mouse chemerin undergoing extensive, dynamic, and tissue-specific proteolytic processing at homologous sites in vivo, similar to human chemerin [27].

Blood samples contain a mix of these forms, with the levels of the different forms depending on the status of the person contributing the sample. Chemerin activation is detected in plasma and adipose tissues from people with obesity undergoing bariatric surgery, and further C-terminal processing occurs during the disposition of chemerin from adipose tissue, resulting in substantial levels of novel degraded forms in plasma that correlate with obesity [26]. Chemerin levels are higher in patients with diabetes, especially those with diabetic complications such as diabetic nephropathy [28,29,30].

Studies using techniques for the direct measurement of insulin action have demonstrated that insulin-stimulated glucose uptake varies several fold in apparently healthy individuals [31,32], and approximately one-third of these individuals are sufficiently IR to be at high risk of developing diabetes. Low-grade inflammation is a feature of individuals with either insulin resistance or diabetes and may contribute to the progression of diabetes [33,34]. A component of this inflammation is increased proteolysis, due to the secretion of enzymes and inactivation of proteolytic inhibitors, thereby leading to more cleavage of circulating cytokines, chemokines, and adipokines including chemerin [23].

Insulin resistance is not routinely diagnosed because there are no routine simple tests available. The insulin suppression test is the gold standard but is complicated, expensive, and has low throughput. Simpler tests can be administered such as the oral glucose tolerance test in which a standard quantity of glucose is administered to a person and blood withdrawn over the next two to three hours for glucose testing; however, this test still requires significant time and expense. A simple blood test would enable people with insulin resistance to be diagnosed and treated.

Chemerin levels in blood have a positive correlation with BMI [35,36] but, to our knowledge, there have been no reports comparing the levels of different chemerin forms in individuals with diabetes and in those without diabetes with different degrees of insulin resistance. We hypothesized that, due to higher inflammation, chemerin levels would be higher and more chemerin activation would occur in individuals with diabetes than in those without diabetes. Furthermore, among individuals without diabetes, those who are IR would have higher levels of chemerin and more chemerin activation than those who are insulin sensitive. To test this hypothesis, we measured the levels of different forms of chemerin in individuals with diabetes and in those without diabetes divided into groups based on their degree of insulin resistance.

## 2. Materials and Methods

### 2.1. Study Participants

The study included individuals who had participated in the studies of insulin resistance and diabetes at Stanford to evaluate the role of insulin resistance in human diseases. The study participants were recruited from the San Francisco Bay Area through print advertisements. All studies were approved by the Stanford Institutional Review Board, and all individuals gave written informed consent to participate in the studies and for use of their data in analyses regarding the role of insulin resistance in human disease. The study participants were in good general health and 18 to 80 years old. The participants with diabetes were either receiving treatment with one or more glucose lowering medications for management of hyperglycemia or had fasting glucose concentration ≥126 mg/dL on more than one occasion. Participants with T2D had FPG levels of 179 ± 27 mg/dL (mean ± SD; range: 150–258 mg/dL).

### 2.2. Insulin Suppression Tests

The degree of insulin resistance was quantified in all participants using a modified version of the insulin suppression test (IST). After an overnight fast, an intravenous catheter was placed in each arm vein. One arm was used for administering a continuous 180-min infusion of octreotide acetate (0.27 µg/m^2^/min), insulin (32 mU/m^2^/min), and glucose (267 mg/m^2^/min). The contralateral arm was used for obtaining blood samples for glucose and insulin measurements. Blood samples were obtained every 30 min until 150 min then every 10 min during the last 30 min of the test to measure the steady-state plasma insulin (SSPI) and glucose (SSPG) concentration. During the IST, the octreotide acetate suppresses endogenous insulin secretion and SSPI concentrations are similar in all individuals. The SSPG concentration provides a direct measure of insulin-stimulated glucose uptake, where a higher SSPG concentration indicates a greater degree of insulin resistance than a lower SSPG concentration. Insulin-stimulated glucose uptake by the IST highly correlates with that by the euglycemic hyperinsulinemic clamp [31,37].

Glucose was measured by the glucose oxidase method and insulin was measured by a radioimmunoassay.

### 2.3. Study Participant Groups

Participants with type 2 diabetes (T2D) composed the T2D group (*n* = 18). Of the 18 patients with T2D, eight were not receiving treatment of glucose lowering medication, whereas seven were being treated with sulfonylurea, two with metformin, and one with thiazolidinedione. As the degree of insulin resistance, quantified by the SSPG concentration, is a continuous phenotypic trait, participants without T2D (*n* = 116) were grouped into SSPG concentration tertiles to categorize them by their degree of insulin resistance. Specifically, participants with SSPG concentration in the lowest tertile (≤91 mg/dL) were defined as insulin sensitive (IS; *n* = 39), those with SSPG concentration in the intermediate tertile (92–199 mg/dL) as intermediate IR (IM; *n* = 38), and those with SSPG concentration in the highest tertile (≥200 mg/dL) as IR (*n* = 39). Thus, four groups were included in the analysis: IS, IM, IR, and T2D.

### 2.4. Samples for Analysis

Blood samples were collected for various measurements after an overnight fast. Following collection, plasma was immediately separated, aliquoted, and stored at −80 Celsius. The samples had undergone one freeze-thaw cycle prior to chemerin measurements.

### 2.5. Chemerin ELISAs, Purification, and Characterization

Total chemerin was determined in plasma samples using R&D systems’ recombinant human chemerin and antibody (R&D Systems, Minneapolis, MN, USA). The generation and purification of recombinant chem163S, chem158K, chem157S, chem156F, chem155A, and their specific antibodies: anti-chem163S, anti-chem158K, anti-chem157S, anti-chem156F, and anti-chem155A were described previously along with the development and validation of specific chemerin ELISAs for these forms [24,26,38]. Recombinant chem144D was produced in mammalian culture (GeneScript USA Inc., Piscataway, NJ, USA) and purified to homogeneity by ion-exchange chromatography [25]. Anti-human chemerin 144D (anti-chem144D) was raised in chickens against the peptide sequence from the C-terminus of chem144D (CLRVQRAGED) conjugated to KLH (Aves Labs, Davis, CA, USA) and anti-chem144D IgY from eggs was purified by affinity chromatography to its cognate peptide bound to Sepharose. The chem144D ELISA consisted of binding the capture antibody, a rat monoclonal anti-human chemerin antibody (4 μg/mL; R&D Systems) in PBS buffer onto 96-well ELISA plates. Nonspecific binding sites were blocked with 1% BSA in PBS for 1 h before sample addition. Purified recombinant Chem144D was used as the standard to construct a calibration curve. Samples and standards were diluted with 1% BSA in PBS and incubated in the wells for 2 h. After washing with 0.05% Tween 20 in PBS, the wells were incubated with specific cognate antibodies (500 ng/mL) in PBS with 1% BSA for 1 h. The wells were washed with 0.05% Tween 20 in PBS before the samples were incubated with peroxidase-conjugated goat anti-chicken IgY antibody (100 ng/mL) in PBS with 1% BSA for 1 h. After washing, tetramethylbenzidine substrate (Alpha Diagnostic International, San Antonio, TX, USA) was incubated for 10 min followed by the addition of Stop Solution (Alpha Diagnostic International), and absorbance at 450 nm was measured. The concentrations of human chemerin forms were calculated from the calibration curves of the purified chemerin standards.

### 2.6. Determination of Chemerin Forms in Plasma Samples from Patients with Insulin Suppression Tests

After thawing, 500–1000 μL of plasma was diluted to 1000 μL with PBS and mixed with 100 μL of heparin-agarose (Sigma, St. Louis, MO, USA) and 22 μL of ×50 Complete Protease Inhibitor (Roche Applied Science, Pleasanton, CA, USA). After washing twice with 500 μL of PBS with Complete Protease Inhibitor, the chemerin bound to heparin-agarose was eluted with 0.8 M NaCl in PBS with Complete Protease inhibitor. The eluates were assayed by ELISAs for total chemerin and specific chemerin forms [24,33]. 

### 2.7. Statistical Analyses

Results are presented as mean ± SEM. Cleaved chemerin forms were calculated by subtracting the value of chemerin 163S forms from total chemerin forms. Degraded chemerin forms were calculated by subtracting the sum of the specific chemerin forms (chem163S, chem158K, chem[157S+156F], and chem155A) from total chemerin forms. Data were analyzed by GraphPad Prism v9.3.1 (GraphPad Software Inc., La Jolla, CA, USA) using two-tailed Student’s *t*-tests with *p* < 0.05 accepted as significant. Correlations were analyzed after polynomial regression analysis using Pearson correlation with two-tailed *p* value. The SPSS regression model (IBM SPSS Software v26, IBM, Armonk, NY, USA) was used to analyze and predict the multi-variable correlation. Multiple regression models were also built in Python using the scikit-learn package [39]. Normality of the distribution of residuals was validated using a normal QQ plot showing that the data fit a normal distribution. Homogeneity of variance of the residuals was checked using a scale-location plot as well as White’s test, both of which revealed that the variance was nonconstant. We acknowledge that some statistical tests may be biased by this heteroscedasticity. Linearity of the residuals was confirmed using a residuals vs. fitted plot. Leverage points were identified using a residuals vs. leverage plot and confirmed to be accurate. Summary statistics for the multiple regression models and classification models were derived using the scikit-learn and statsmodels packages [40]. For the classification models, we used an 80/20 split for the training/testing set. Four common classification algorithms (decision tree, random forest, naive Bayes, and k-nearest neighbors) were used to construct the models.

## 3. Results

### 3.1. Development of an ELISA Specific for chem144D

When chemerin was purified from plasma obtained from bariatric surgery patients for analysis by mass spectrometry, a novel degraded form of chemerin, chem144D, was detected [26]. In order to routinely determine the levels of chem144D, we developed a specific ELISA for chem144D using chicken anti-chem144D, prepared by immunization of hens with a peptide representing the C-terminal sequence of chem144D. The specificity of the affinity-purified anti-chem144D IgY was demonstrated by ELISA in which anti-chem144D IgY only recognized its target chemerin protein but not the other five chemerin forms tested, with a lower limit of detection of 0.32 ng/mL (Figure 1). Confirmation of the specificity of the chem144D ELISA was obtained by including the cognate chem144D peptide in the assay, which eliminated any response to the chem144D protein. The earlier mass spectrometry data identifying chem144D as a degradation product was verified by the positive signal found when plasma was analyzed with this ELISA.

### 3.2. Levels of Different Chemerin Forms in Plasma

Plasma samples were collected from 134 participants who had previously had their degree of insulin resistance determined by the SSPG concentration during the IST and were divided into four groups as described in Section 2.3. As noted above, there were 18 participants with T2D whose medications are listed in Section 2.3 and 116 without T2D. The demographic characteristics of the four groups were similar, but the fasting plasma glucose (FPG) continuously increased across the groups (Table 1).

The level of the different forms of plasma chemerin present in these samples was determined by the total chemerin ELISA and five specific ELISAs for chem163S, chem158K, chem157S+156F, chem155A, and chem144D (Figure 2 and Table 2). When the mean levels of total chemerin were compared among the four groups, we found that the chemerin levels of IS and IM participants were lower (60 ± 2.8 ng/mL and 56 ± 2.2 ng/mL, respectively) than those of either IR participants (70 ± 3.3 ng/mL; *p* = 0.0335 and *p* = 0.001) or T2D participants (93 ± 6.1 ng/mL, *p* < 0.0001; Figure 3 and Table 3).

### 3.3. Proteolytic Processing of Chemerin in Plasma

To evaluate the amount of proteolytic processing of chemerin occurring in different groups, we measured the concentrations of different chemerin forms in plasma from our study participants. The mean levels of chem163S, prochemerin, were 29.6 ± 1.4 ng/mL in IS participants, 35.5 ± 2.0 ng/mL in IM participants, 30.7 ± 1.7 ng/mL in IR participants, and 29.3 ± 3.0 ng/mL in T2D participants; while levels of partially active chem158K were higher in IR and T2D participants (7.3 ± 0.4 ng/mL and 7.3 ± 0.6 ng/mL, respectively) than in either IM (6.0 ± 0.2 ng/mL, *p* = 0.0048 and *p* = 0.0069, respectively) or IS participants (6.0 ± 0.3 ng/mL, *p* = 0.0121 and *p* = 0.0272). The mean levels of chem157S and chem156F, the active chemerin forms, were 24.4 ± 2.5 ng/mL in IR participants, 20.4 ± 2.3 ng/mL in T2D participants, 16.7 ± 2.0 ng/mL (*p* = 0.0204) in IM participants, and 16.9 ± 2.3 ng/mL (*p* = 0.0322) in IS participants. While levels of chem155A, an inactive chemerin form, were similar in IS (1.4 ± 0.1 ng/mL), IM (1.5 ± 0.0 ng/mL), and IR (1.6 ± 0.1 ng/mL) participants, its level in T2D participants was lower (1.2 ± 0.1 ng/mL; ns, *p* = 0.0081, and *p* = 0.0342, respectively). Chem144D, one of the degraded chemerin forms, was significantly higher in IR (1.8 ± 0.1 ng/mL) than in IM (1.4 ± 0.1 ng/mL, *p* < 0.0001) and IS (1.4 ± 0.2 ng/mL, ns) participants (Figure 4A). Chem144D level was not determined in T2D participants because we had not yet validated the chem144D specific ELISA. Calculating the average of the different chemerin forms in the four groups showed that the T2D participants contained the most proteolytically cleaved chemerin, and IR participants contained the second most proteolytically cleaved chemerin (Figure 4B).

### 3.4. Levels of Cleaved Chemerin in Plasma

Cleaved chemerin, defined as all chemerin forms smaller than chem163S, is calculated by subtracting the value of chem163 from total chemerin and includes all proteolytically cleaved chemerin forms shorter than the precursor, chem163S. The level of cleaved chemerin is a measure of the overall proteolysis of chemerin that has occurred in the study participants. The mean levels of cleaved chemerin were significantly higher in T2D participants (64 ± 7.7 ng/mL) than in IS (31 ± 3.1 ng/mL, *p* < 0.0001), IM (21 ± 1.6 ng/mL, *p* < 0.0001), and IR (39 ± 3.7 ng/mL, *p* = 0.0016) participants (Figure 5A and Table 3).

### 3.5. Levels of Degraded Chemerin in Plasma

Degraded chemerin is defined as all chemerin forms smaller than chem155A and possess no activity themselves and they cannot be subsequently activated. Degraded chemerin, calculated by subtracting the sum of the chemerin forms (chem163S, chem158K, chem[157S+156F], and chem155A) from total chemerin, was also significantly higher in T2D (36 ± 7.4 ng/mL) participants than in IS (10 ± 2.7 ng/mL, *p* = 0.0001), IM (2.4 ± 1.1 ng/mL, *p* < 0.0001), and IR (14 ± 2.9 ng/mL, *p* = 0.0012) participants (Figure 5B and Table 3). These data show that there was more proteolytic processing of chemerin resulting in higher levels of both cleaved and degraded chemerin in T2D participants, followed by in IR participants, than in IS and IM participants.

### 3.6. Relationship between Chemerin Levels and SSPG

Circulating chemerin levels are increased in patients with diabetes, so we investigated the relationship between SSPG concentration and the levels of different chemerin forms. When chemerin levels were compared to SSPG concentration by regression analysis, total, cleaved, and degraded chemerin levels demonstrated a robust increase with increasing SSPG concentration with adjusted R squared values of 0.311, 0.322, and 0.345, respectively (Figure 6A–C, Table 4). In contrast, the individual species of chemerin (chem163S, chem158K, chem157S, chem156F, chem155A, and chem144D) did not exhibit the same positive correlation. When the participants with T2D were analyzed separately from the participants without T2D, both sets exhibited significant correlations of SSPG concentration with total, cleaved, and degraded chemerin.

### 3.7. Relationship between Chemerin Levels and BMI

As obesity has been shown to affect the levels of both chemerin and its cleavage [26,28,29,30], we analyzed by regression analysis the relationship between BMI and chemerin levels. We found that total, cleaved, and degraded chemerin had a positive correlation with BMI with adjusted R squared values of 0.089, 0.066, and 0.034, respectively. (Figure 6D–F; Table 4). These positive correlations were statistically significant but, because of their small size, almost certainly not biologically relevant. The individual chemerin species (chem163S, chem158K, chem157S, chem156F, chem155A, and chem144D), however, did not possess a clear correlation with BMI. If the T2D group was analyzed separately from the group without T2D, then both the participants with T2D and those without still exhibited the correlations of BMI with total, cleaved, and degraded chemerin. The correlation of BMI with the levels of the various chemerins was less robust than its correlation with SSPG concentration.

### 3.8. SSPG and BMI Are Confounders for Prediction of Chemerin Levels

Circulating chemerin levels are influenced by both diabetes and obesity but there is no evidence, to our knowledge, whether they are independent variables in determining chemerin levels. To investigate this question, we analyzed by multivariate polynomial analysis if BMI was a confounder of the correlation between SSPG concentration and total, cleaved, and degraded chemerin levels. We found that that including BMI with SSPG concentration in the multivariate analysis did not lead to a large improvement in adjusted R squared compared to the values for SSPG concentration alone (Table 4). This was confirmed by the small changes in the Akaike information criterion when BMI was included with SSPG concentration. This suggests that SSPG concentration and BMI are confounders in predictions of the levels of total, cleaved, and degraded chemerin.

### 3.9. Use of Chemerin Levels Improves Diagnosis of Insulin Resistance

We used correlation analysis to evaluate if the chemerin measurements in participant without T2D correlated with SSPG concentration. Using adjusted R^2^ and AIC (Akaike information criterion), in which higher R^2^ and lower AIC values mean a stronger correlation and a better model fit, SSPG concentration correlated best with BMI, then from higher to lower correlation with FPG, cleaved chemerin, total chemerin, and degraded chemerin (Table 5).

We constructed multi-variable regression models to predict SSPG concentration values in participants without T2D using the clinical markers, BMI, and FPG, in combination with different chemerin measurements to determine which models performed best. We found that including total chemerin, cleaved chemerin, or degraded chemerin as variables improved the accuracy of SSPG prediction compared to models using only BMI and FPG (Table 6). Similarly, including total chemerin, cleaved chemerin, or degraded chemerin as variables improved the accuracy of SSPG concentration prediction compared to models using only BMI or FPG (Table 5 and Table 6). The AIC and adjusted R squared for the multi-variable regression models showed a better correlation with SSPG concentration than the single-variable correlations. The model giving the best predictive power for SSPG concentration employed BMI, FPG, and one of total chemerin, cleaved chemerin, or degraded chemerin as variables and was an improvement over BMI and FPG alone.

Based on that improvement, we constructed classification models to evaluate if including total chemerin or cleaved chemerin as part of the evaluation of insulin resistance in participants without T2D would be an improvement over use of BMI and FPG alone. To do that, we analyzed the performance of predicting insulin resistance, defined as SSPG > 200 mg/dL using BMI and FPG in models with or without including total chemerin or cleaved chemerin measurements. Two models were constructed, and their receiver operating characteristic (ROC) curves were compared using naive Bayesian analysis. The ROC analysis with BMI and FPG that included degraded, cleaved chemerin, or total chemerin showed a clear improvement in AUC (area under curve) over the ROC analysis without degraded, cleaved chemerin, or total chemerin (Figure 7). Similarly, addition of total, cleaved, or degraded chemerin to FPG showed an improvement in ROC AUC over FPG alone (Appendix A). Comparison of the ROC AUCs shows that performance is better when both BMI and FPG are included along with a chemerin parameter (Appendix A).

We confirmed this result by constructing the ROCs using other models, all of which showed an increased AUC (Appendix A) demonstrating that including cleaved chemerin data improved the models’ ability to correctly classify individuals as IR.

In these models, 80% of the individuals were assigned randomly to the training set and 20% to the test set. To show that the improvement in the ROC AUC by including the degraded chemerin, cleaved chemerin, or total chemerin value was robust, the models were run four times with a different randomization between training and test sets for each iteration. Each repeat gave the same result that including degraded chemerin, cleaved chemerin, or total chemerin improved the ROC AUC.

## 4. Discussion

Chemerin is secreted as a precursor (prochemerin) with low biological activity that terminates in humans at amino acid serine 163 (chem163S) [23]. Prochemerin is converted into a full agonist for chem1 and chem2 by truncation of the last six amino acids at its C-terminus by proteases belonging to the coagulation, fibrinolytic, and inflammatory cascades [41,42,43,44]. The most active form of human chemerin, chem157S, can be generated either by direct cleavage of prochemerin by neutrophil-derived serine proteases (elastase or cathepsin G) or tissue-kallikrein [45], or alternatively by sequential cleavages by clotting factor FXIa or by plasmin to form chem158K, which has modest activity, followed by the removal of the C-terminal lysine by carboxypeptidase N (CPN) or carboxypeptidase B2 (CPB2, also termed thrombin-activatable fibrinolysis inhibitor) producing chem157S [43]. Chymase was shown to be capable of cleaving chem163S to chem156F, a form of chemerin partially active on chem1, which is more active than its precursor, chem163S, but very much less potent than chem157S on chem1 [22,24]. Further enzymatic proteolysis also inactivates bioactive chemerin. Neutrophil-derived protease 3, mast cell chymase [46], and angiotensin-converting enzymes [47] can all convert active chemerin into inactive derivatives including chem155A and smaller forms. Thus, precise proteolytic processing is a key regulatory mechanism that determines both systemic and local concentrations of bioactive chemerin. However, chemerin levels as measured by currently commercially available chemerin ELISAs do not distinguish between the various forms of active and inactive chemerin and have not been validated for their response to different chemerin forms. Both of these issues pose significant limitations in understanding the role of chemerin.

In a prospective cohort study, chemerin levels were found to predict the risk of cardiovascular disease independently of other risk factors, plus there was a strong positive association with T2D [48]. High chemerin levels correlate with increased all-cause mortality, primarily via a raised risk of cancer [49]. In patients with obesity, chemerin levels are increased and more activation of chemerin occurs [7,26,50,51]. Chemerin levels are also increased in patients with metabolic syndrome and both type 1 diabetes and T2D [35,52,53,54]. Diabetic kidney disease was associated with higher levels of serum chemerin [55].

Circulating levels of chemerin are increased in patients with obesity undergoing bariatric surgery [26], but that increase is not due to significant differences in levels of the precursor, chem163S. In this study, we found similar results in patients with insulin resistance and T2D with the total level of chemerin highest in the T2D group, and the second highest in the IR group, but the levels of chem163S were similar (Figure 3 and Figure 4A). This implies that the difference in total chemerin is due to increased levels of circulating chemerin that had been proteolytically cleaved. When we investigated the proteolytically cleaved forms, the T2D group had higher levels of them than the IR group and than either of the other two groups. In addition, the T2D and IR groups had increased levels of degraded chemerin. Taken together, these results suggest that there was more ongoing proteolysis in the T2D group, followed by IR group, than the IS or IM groups, probably due to the inflammation associated with the underlying diabetes confirming our original hypothesis [56,57]. These data are consistent with the hypothesis that an individual progresses from being insulin sensitive to IR and finally to overt T2D; the increased inflammation results in more production of chemerin as well as greater proteolysis of it [58,59].

To our knowledge, this is the first study on chemerin levels and forms that has used the IST, a direct method for quantifying the degree of insulin resistance [60,61]. Although the IST is the gold-standard for determining insulin resistance, the procedure is time-consuming, labor intensive, expensive, and, therefore, it is impractical to apply it in large epidemiological studies and burdensome in the clinical care setting.

As obesity and insulin resistance are highly correlated [62,63], we also analyzed the effect of BMI on chemerin levels and processing. Analysis of the relationship of BMI with total cleaved and degraded chemerin levels showed that BMI also strongly influenced chemerin levels, and multivariate analysis suggested that BMI is a confounder for the association of total cleaved and degraded chemerin levels with SSPG concentration. This probably represents a single metabolic dysfunction influencing chemerin levels and cleavage. Insulin resistance causes an inflammatory state in which the innate immune system is activated [64]. Both liver and adipose tissue respond to insulin resistance and obesity by increasing chemerin production and, in the case of adipose tissue, its activation [26,65,66]. The association of total cleaved and degraded chemerin levels with glucose levels and BMI could result from chemerin modulating energy metabolism, energy metabolism regulating chemerin, or both being affected by a third factor.

Chemerin is key in both obesity and insulin resistance in chemerin deficient mice, but the available data do not delineate the mechanistic relationship between chemerin, obesity, and insulin resistance [10,20,21,67]. This is supported by data from mice that are deficient in one of the two signaling chemerin receptors, chem1 or chem2 [19,68]. Chem1 deficiency, however, did not affect insulin resistance in one report, but elsewhere the data show that insulin resistance and obesity are affected by chem1 [19,68,69]. Obese chem2 deficient mice develop worse insulin resistance than WT mice showing the critical role of chem2 [12]. To our knowledge, none of these models has explored the relationship between the various factors under investigation. Thus, all three of the causative models are consistent with the currently available murine in vivo data. In addition, there may be further feedback loops between glucose, weight gain, and chemerin metabolism.

The data presented here suggest that in this cohort of participants without T2D the use of chemerin levels can contribute to improved diagnosis of insulin resistance. Although data are presented on differences in levels of several of the chemerin forms, the greatest improvement in diagnostic power was achieved with incorporating the levels of cleaved chemerin or degraded along with FPG and BMI. To determine cleaved chemerin, two ELISAs are needed, one for total chemerin and the other for prochemerin (chem163S). This combination offers the greatest improvement in diagnosis with the fewest assays. While incorporating the levels of total chemerin improves diagnostic power similarly, the fact that the increased level of chemerin is due to cleaved chemerin was not revealed. We investigated whether our best models perform differently in the different groups, and we found that the model was significantly more accurate for the IR and T2D groups than either the IS or IM group (Appendix A), indicating that chemerin measurements are correlated best with SSPG concentration when an individual is resistant to insulin.

Treatment of patients with obesity, insulin resistance, and T2D is rapidly changing with the introduction of glucagon-like peptide-1 (GLP-1) receptor agonist treatments such as liraglutide and semaglutide [70,71,72]. To our knowledge, there have not been any studies investigating effects of these therapies on chemerin levels and activation. Chemerin expression in the liver and chemerin serum levels are reduced in a rat model of insulin resistance induced by a high fat diet when the animals are treated with liraglutide [73]. This suggests that GLP-1 down-regulates chemerin. Conversely, chemerin may down-regulate GLP-1 expression and secretion [74]. GLP-1 and chemerin also have opposing effects on macrophage polarization with GLP-1 promoting M2 formation and chemerin inhibiting it [75,76,77]. Taken together, this suggests a mutual feedback control of GLP-1 and chemerin production which would have implications for overall control of energy balance and inflammation.

The next steps will be to increase the group size and to confirm these data in independent cohorts. In this study, we were unable to analyze the relationship with Hb A1c as the data were not available in the records of the determinations of SSPG concentration so that should also be included as the focus of future studies. Studies on chemerin levels and activation in patients treated with GLP-1 receptor agonists would be informative.

In summary, this study showed that T2D and IR individuals had increased levels of cleaved and degraded chemerin compared to individuals who were IS and IM and those measurements could be useful for understanding the cardiometabolic risk associated with T2D and insulin resistance.

## Figures and Tables

**Figure 1 biomedicines-12-00924-f001:**
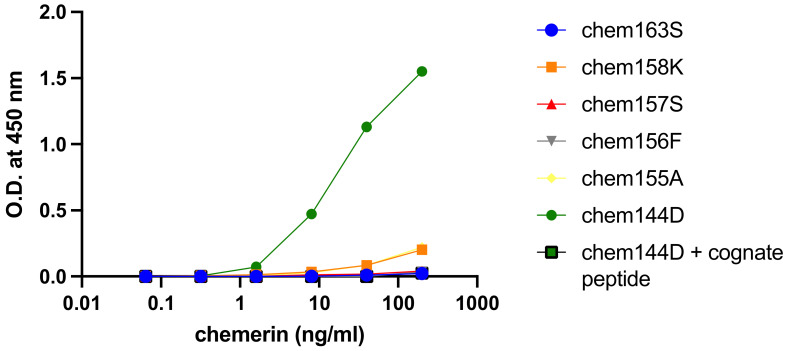
Characterization of specific antibody against recombinant chem144D. Recombinant chem163S (blue circle), chem158K (orange square), chem157S (red triangle), chem156F (gray triangle), chem155A (yellow diamond), and chem144D (green circle) by anti-chem144D IgY were detected using specific chem144D ELISA as described in the “Materials and Methods” Section 2.5. Only recombinant chem144D is detected. Cognate peptide (green square) competes in the ELISA.

**Figure 2 biomedicines-12-00924-f002:**
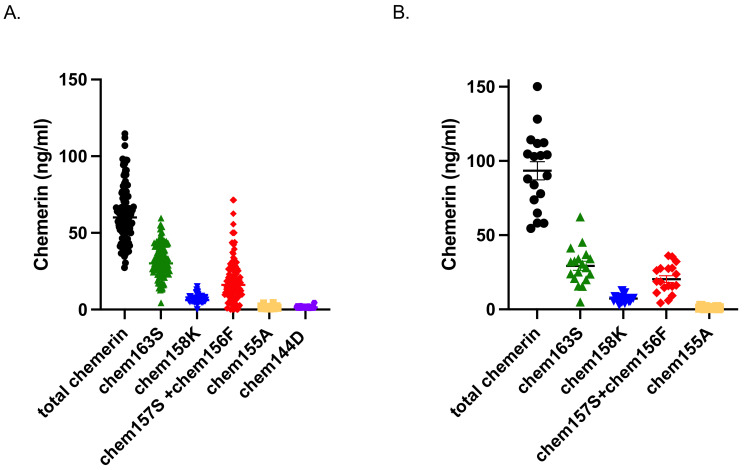
Levels of chemerin forms in plasma from the participants with and without T2D included in the study. Total chemerin, Chem163S, Chem158K, Chem157S, and Chem156F, Chem155A, and Chem144D levels in human plasma from 116 participants without T2D (**A**), and 18 participants with T2D (**B**) were determined using total chemerin ELISA and specific chemerin ELISAs as described in the “Materials and Methods” Section 2.5.

**Figure 3 biomedicines-12-00924-f003:**
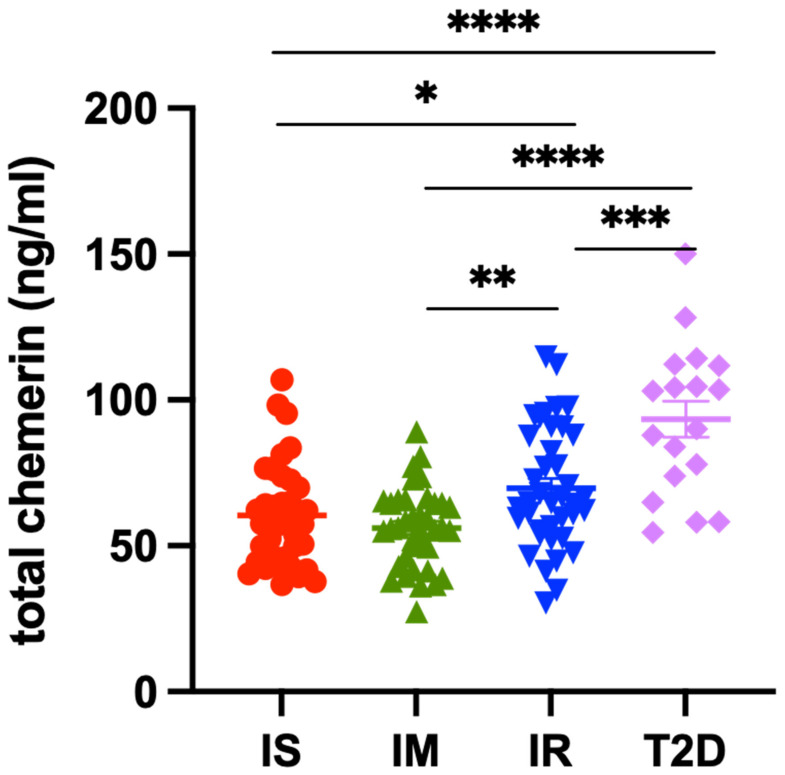
Levels of plasma total chemerin in individuals who were insulin sensitive (IS, SSPG ≤ 91 mg/dL, *n* = 39), intermediate (IM, SSPG 92–199 mg/dL, *n* = 38), insulin resistant (IR, SSPG ≥ 200 mg/dL, *n* = 39), or had diabetes (T2D, *n* = 18). In the IS, IM, IR, and T2D groups, total chemerin levels were determined using the total chemerin ELISA described in the “Materials and Methods” Section 2.5. Colored horizontal thick lines show the mean and thin lines ± SEM. *: <0.05, **: <0.01. ***: <0.001, ****: <0.0001.

**Figure 4 biomedicines-12-00924-f004:**
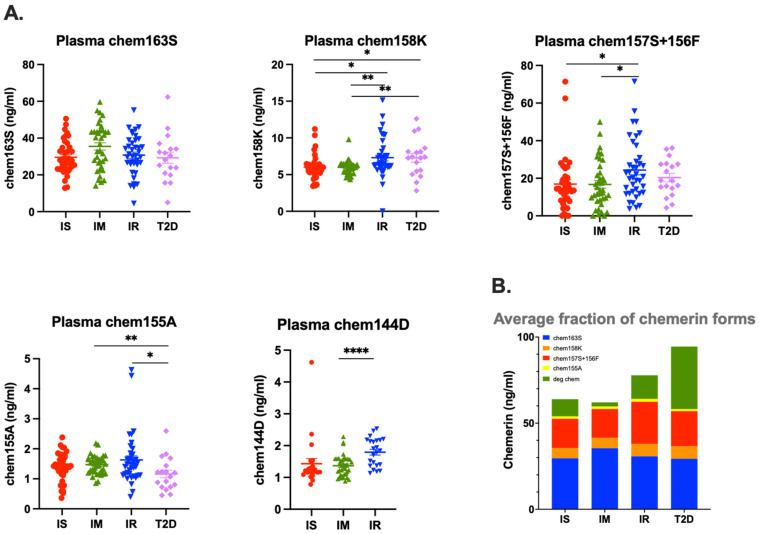
Levels of plasma specific chemerin in individuals who were insulin sensitive (IS, SSPG ≤ 91 mg/dL, *n* = 39), intermediate IR (IM, SSPG 92–199 mg/dL, *n* = 38), insulin resistant (IR, SSPG > 200 mg/dL *n* = 39), or had diabetes (T2D, *n* = 18). (**A**) Chem163S, Chem158K, Chem157S and Chem156F, Chem155A, and Chem144D levels in plasma of IS, IM, IR, and T2D participants were determined using specific chemerin ELISAs as described in the “Materials and Methods” Section 2.5. Colored horizontal thick lines show the mean and thin lines ± SEM. *: <0.05, **: <0.01. ****: <0.0001. (**B**) The average fraction of the different chemerin forms in the four study groups are displayed.

**Figure 5 biomedicines-12-00924-f005:**
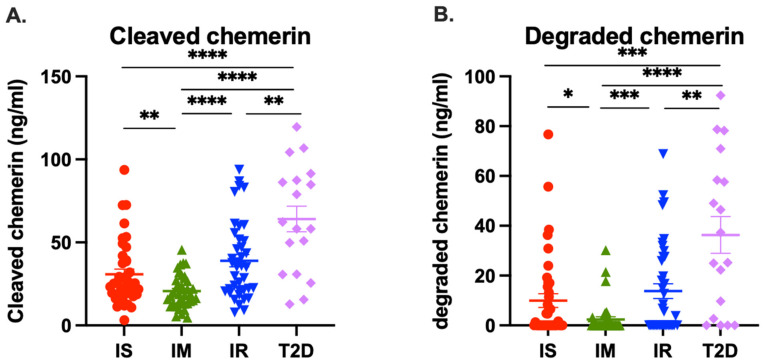
Levels of plasma cleaved and degraded chemerin in samples from individuals who are insulin sensitive (IS, SSPG ≤ 91 mg/dL, *n* = 39), intermediate IR (IM, SSPG 92–199 mg/dL, *n* = 38), insulin resistant (IR, SSPG ≥ 200 mg/dL, *n* = 38), or had diabetes (T2D, *n* = 18). Cleaved chemerin (**A**) and degraded chemerin (**B**) levels in plasma of IS, IM, IR, and T2D participants were determined as described in the “Materials and Methods” Section 2.5. Horizontal lines show the mean (thick line) ± SEM (thin lines). *: <0.05, **: <0.01. ***: <0.001, ****: <0.0001.

**Figure 6 biomedicines-12-00924-f006:**
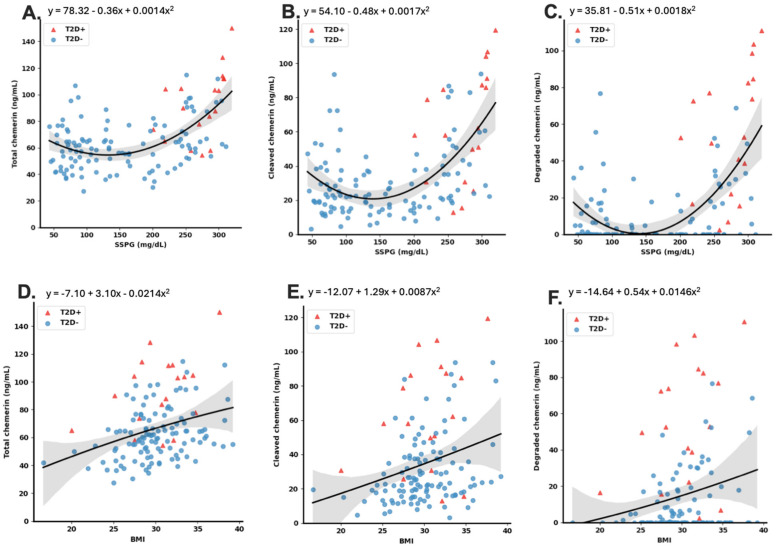
Correlations of levels of total, cleaved, and degraded chemerin with SSPG concentration and BMI. Correlations included all samples and were fitted by regression analysis to exponential growth curves as described in the Materials and Methods Section 2.7. The regression is represented by the solid line with its equation shown and the grey shaded area represents the 95% confidence intervals: (**A**) SSPG concentration vs. total chemerin (**B**) SSPG concentration vs. cleaved chemerin (**C**) SSPG concentration vs. degraded chemerin (**D**) BMI vs. total chemerin (**E**) BMI vs. cleaved chemerin (**F**) BMI vs. degraded chemerin. T2D^+^: participants with diabetes, T2D^−^: participants without diabetes.

**Figure 7 biomedicines-12-00924-f007:**
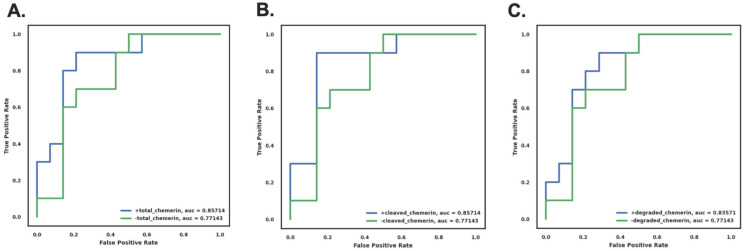
ROC curve of naive Bayes classification model using BMI and FPG in combination with or without total chemerin (**A**), with or without cleaved chemerin (**B**) and with or without degraded chemerin (**C**) in individuals without T2D. Classification models using the naive Bayesian algorithm were constructed as described in the “Materials and Methods” Section 2.7. The ROC curves with total chemerin, cleaved chemerin, or degraded chemerin (displayed in blue) show an improvement in AUC (0.86 ± 0.08, 0.86 ± 0.08 and 0.84 ± 0.09 for total chemerin, cleaved chemerin, or degraded chemerin, respectively) over that of ROC without degraded, cleaved, or total chemerin (displayed in green) (0.77 ± 0.1).

**Table 1 biomedicines-12-00924-t001:** Demographic characteristics of the groups of individuals who were insulin sensitive (IS, SSPG ≤ 91 mg/dL), intermediate IR (IM, SSPG 92–199 mg/dL), IR (SSPG ≥ 200 mg/dL), or had diabetes (T2D). Data shown as mean ± SEM. Data were analyzed using unpaired *t*-tests with *p* < 0.05 accepted as significant. SSPG, steady-state plasma glucose; BMI, body mass index; FPG, fasting plasma glucose; and ns, nonsignificant.

Mean ± SEM	IS (*n* = 39)	IM (*n* = 38)	IR (*n* = 39)	T2D (*n* = 18)	*p* Value	*p* Value	*p* Value	*p* Value	*p* Value	*p* Value
					IS vs. IM	IS vs. IR	IS vs. T2D	IM vs. IR	IM vs. T2D	IR vs. T2D
Age	51.4 ± 1.3	48.9 ± 1.6	54.6 ± 1.5	54.4 ± 2.0	ns	ns	ns	0.0133	0.0458	ns
Gender	M-15, F-24	M-12, F-26	M-18, F-21	M-10, F-8	ns	ns	ns	ns	ns	ns
	38.5% male	31.6% male	46.2% male	55.6% male						
SSPG (mg/dL)	70.0 ± 2.2	137.6 ± 5.1	250.4 ± 4.9	274.4 ± 8.4	<0.0001	<0.0001	<0.0001	<0.0001	<0.0001	0.0122
BMI	28.5 ± 0.6	29.5 ± 0.6	32.1 ± 0.5	30.4 ± 0.9	ns	<0.0001	ns	0.0021	ns	ns
FPG (mg/dL)	91.4 ± 1.4	94.3 ± 1.3	102.9 ± 1.2	179.1 ± 6.4	ns	<0.0001	<0.0001	<0.0001	<0.0001	<0.0001

**Table 2 biomedicines-12-00924-t002:** Total chemerin and specific chemerin level profile in plasma from participants with diabetes (T2D+) or without diabetes (T2D-). Data are summarized from Figure 2 and given as mean ± SEM. N.D.: not determined.

Mean ± SEM (ng/mL)	Total Chemerin	chem163S	chem158K	Chem[157S+156F]	chem155A	chem144D
T2D+	93.42 ± 6.15	29.31 ± 3.03	7.29 ± 0.6	20.37 ± 2.28	1.16 ± 0.13	N.D.
T2D-	62.14 ± 1.69	31.90 ± 1.01	6.44 ± 0.18	19.35 ± 1.35	1.50 ± 0.05	1.50 ± 0.06

**Table 3 biomedicines-12-00924-t003:** Total chemerin, specific chemerin, cleaved chemerin, and degraded chemerin levels in plasma of individuals who were in the IS, IM, IR, or T2D groups. Data are summarized from Figure 3 and Figure 4 and are given as mean ± SEM. Data were analyzed using unpaired *t*-tests with *p* < 0.05 accepted as significant. Percent (%) of total chemerin for each chemerin form is shown. N.D.: not determined; ns: not significant.

Mean ± SEM (ng/mL)	IS (*n* = 39)	IM (*n* = 38)	IR (*n* = 39)	DM (*n* = 18)	*p* Value	*p* Value	*p* Value	*p* Value	*p* Value	*p* Value
(% of Total Chemerin)					IS vs. IM	IS vs. IR	IS vs. DM	IM vs. IR	IM vs. DM	IR vs. DM
Total chemerin	60.4 ± 2.8	56.2 ± 2.2	69.7 ± 3.3	93.4 ± 6.1	ns	0.0335	<0.0001	0.001	<0.0001	0.0005
chem163S	29.6 ± 1.4	35.5 ± 2.0	30.7 ± 1.7	29.3 ± 3.0	0.0177	ns	ns	ns	ns	ns
	−49.00%	−63.20%	−44.00%	−31.40%						
chem158K	6.0 ± 0.3	6.0 ± 0.2	7.3 ± 0.4	7.3 ± 0.6	ns	0.0121	0.0272	0.0048	0.0069	ns
	−9.90%	−10.70%	−10.50%	−7.80%						
chem157S+156F	16.9 ± 2.3	16.7 ± 2.0	24.4 ± 2.5	20.4 ± 2.3	ns	0.0322	ns	0.0204	ns	ns
	−28%	−29.70%	−35%	−21.80%						
chem155A	1.4 ± 0.1	1.5 ± 0.0	1.6 ± 0.1	1.2 ± 0.1	ns	ns	ns	ns	0.0081	0.0342
	−2.30%	−2.70%	−2.30%	−1.30%						
chem144D	1.4 ± 0.2	1.4 ± 0.1	1.8 ± 0.1	ND	ns	ns	ND	<0.0001	ND	ND
	−2.30%	−2.50%	−2.60%							
Cleaved chemerin	30.8 ± 3.1	20.7 ± 1.6	39.0 ± 3.7	64.1 ± 7.7	0.0054	ns	<0.0001	<0.0001	<0.0001	0.0016
	−51%	−36.80%	−56%	−68.60%						
Degraded chemerin	10.0 ± 2.7	2.4 ± 1.1	13.8 ± 2.9	36.3 ± 7.4	0.0127	ns	0.0001	0.0006	<0.0001	0.0012

**Table 4 biomedicines-12-00924-t004:** Correlations of SSPG concentration and BMI with total, cleaved, and degraded chemerin in both participants with T2D and those without. Correlation analyses with SSPG concentration and BMI were carried out individually in the first two rows and together using multivariate analysis in the last row as described in the “Materials and Methods” Section 2.7. AIC: Akaike information criterion.

	AIC	AIC	AIC	Adjusted R Squared	Adjusted R Squared	Adjusted R Squared
				(*p* Value)	(*p* Value)	(*p* Value)
	Total Chemerin	Cleaved Chemerin	Degraded Chemerin	Total Chemerin	Cleaved Chemerin	Degraded Chemerin
SSPG	1163	1191	1188	0.311	0.322	0.345
				<0.0001	<0.0001	<0.0001
BMI	1200	1233	1240	0.089	0.066	0.034
				<0.0001	<0.0001	<0.0001
SSPG + BMI	1158	1190	1186	0.348	0.341	0.368
				<0.0001	<0.0001	<0.0001

**Table 5 biomedicines-12-00924-t005:** Different models for individual analyte correlations with SSPG concentration in participants without T2D. Polynomial correlation analyses were obtained as described in the “Materials and Methods” Section 2.7. AIC: Akaike information criterion.

Model	AIC	Adjusted R Squared
SSPG/BMI	618	0.214
SSPG/FPG	823	0.214
SSPG/cleaved chemerin	1005	0.158
SSPG/total chemerin	990	0.122
SSPG/degraded chemerin	956	0.113

**Table 6 biomedicines-12-00924-t006:** Multi-variable polynomial regression models to predict participant SSPG concentration using FPG and BMI in combination with different chemerin measurements in participants without T2D. Multi-variable regression models were constructed as described in the “Materials and Methods” Section 2.7. AIC: Akaike information criterion.

Model	AIC	Adjusted R Squared
SSPG/FPG/BMI/total chemerin	1306	0.341
SSPG/FPG/BMI/cleaved chemerin	1305	0.343
SSPG/FPG/BMI/degraded chemerin	1296	0.395
SSPG/FPG/BMI	1306	0.318
SSPG/FPG/total chemerin	1316	0.256
SSPG/FPG/cleaved chemerin	1315	0.262
SSPG/FPG/degraded chemerin	1310	0.292
SSPG/BMI/total chemerin	1319	0.237
SSPG/BMI/cleaved chemerin	1323	0.209
SSPG/BMI/degraded chemerin	1321	0.224

## Data Availability

Raw data for this study are available by application to the corresponding authors.

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
