# Peer review of "Chemerin in Participants with or without Insulin Resistance and Diabetes"

_biomedicines, 2024, doi:10.3390/biomedicines12040924_

Round 1

Reviewer 1 Report

Comments and Suggestions for Authors

In the present study, Zhao L. et al. reported the relationship between plasma different chemerin forms and insulin resistance measured by SSPG. The findings are potentially interesting, but there are several questions that deflate the value of the present study.

Major criticisms

Line 196: The authors analyzed the data of type 2 diabetic patients with high SSPG value. However, they just describe as type 2 diabetes in other parts (e.g. Table 1). Analysis of type 2 diabetic patients only having high insulin resistance might mislead the results. Please check the data of all T2DM.

Table 3: Subjects with intermediate insulin resistance (92<SSPG<200) have the lowest plasma cleaved/degraded chemerin levels. Why? What is the biological meaning? Chemerin levels might be useful to discriminate subjects having very strong insulin resistance, but NOT low/intermediate IR. I suggest that the authors analyze the data from this point.

Table 6: No label of A, B, C. The authors showed the R squared values only. How about the 95% CI? Clearly, the regression curve is not linear. What is the polynominal expression? R2 squared values of 0.07-0.15 seem to be almost no meaning.

Line 471: FPG is not used to estimate the insulin resistance. The authors should use more suitable values, such as fasting insulin levels, for the multi-variable regression analysis (Table 6). Does Total/cleaved chemin improve the AUC  of ROC curve when BMI and F-IRI were used in the analysis?.

Author Response

Reviewer 1

In the present study, Zhao L. et al. reported the relationship between plasma different chemerin forms and insulin resistance measured by SSPG. The findings are potentially interesting, but there are several questions that deflate the value of the present study.

 The authors appreciate the reviewer’s careful reading of our manuscript and have answered the points below.

Major criticisms

  • Line 196: The authors analyzed the data of type 2 diabetic patients with high SSPG value. However, they just describe as type 2 diabetes in other parts (e.g. Table 1). Analysis of type 2 diabetic patients only having high insulin resistance might mislead the results. Please check the data of all T2DM.

Patients with type 2 diabetes are insulin resistant as a group as originally shown by our research group (Shen et al, 1970, JCI PMID: 5480843). Consistent with that, all of the participants in this study with the diagnosis of type 2 diabetes were insulin resistant with SSPG concentration >200 mg/dL.

We have revised the sentence to clarify that point so that it now reads: “Our study included 18 participants with T2D and 116 without T2D. Participants without T2D were divided into 3 groups based on the SSPG concentration tertiles. Participants in the bottom tertile (SSPG concentration <=91 mg/dL) were classified as insulin sensitive (IS, n=39); those in the middle tertile (SSPG concentration 92 – 199 mg/dL) were classified as intermediate (IM, n=38); and those in the top tertile (SSPG concentration >=200 mg/dL) were classified as insulin resistant (IR, n=39). By this classification, all participants with T2D (n=18) were insulin resistant, indicated by SSPG concentration >200 mg/dL.”

  • Table 3: Subjects with intermediate insulin resistance (92<SSPG<200) have the lowest plasma cleaved/degraded chemerin levels. Why? What is the biological meaning? Chemerin levels might be useful to discriminate subjects having very strong insulin resistance, but NOT low/intermediate IR. I suggest that the authors analyze the data from this point.

The determination that the lowest levels of cleaved and degraded chemerin were found in the group with intermediate insulin resistance was not expected and we do not have a mechanistic explanation for this surprising result. We agree that chemerin levels are not a useful discriminator for individuals with low and intermediate IR.

  • Table 6: No label of A, B, C. The authors showed the R squared values only. How about the 95% CI? Clearly, the regression curve is not linear. What is the polynominal expression? R2 squared values of 0.07-0.15 seem to be almost no meaning.

Thanks for pointing out the lack of labels, which has now been rectified. The reviewer is correct that the analysis that we chose to use is polynomial regression analysis. The polynomial equations are now shown on the graphs. We have repeated all of the correlation analyses and rederived the models using polynomial regression, and the revised values are in Tables 4, 5 and S1. The 95% CIs have also been added to the graphs (figure 6). We agree that the low adjusted R squared values for the correlations with BMI are probably not biologically meaningful. A new sentence has been added to the text in section 3.7 to emphasize this point: “These positive correlations were statistically significant but, because of their small size, not biologically relevant.”

  • Line 471: FPG is not used to estimate the insulin resistance. The authors should use more suitable values, such as fasting insulin levels, for the multi-variable regression analysis (Table 6). Does Total/cleaved chemin improve the AUC of ROC curve when BMI and F-IRI were used in the analysis?

We agree that FPG is not an estimate of insulin resistance; however, fasting insulin is a suitable but surrogate estimate of insulin resistance.  In our study, we do not have fasting insulin measurements available, but we used SSPG concentration – a direct measure of insulin resistance obtained during the insulin suppression test.

Reviewer 2 Report

Comments and Suggestions for Authors

This manuscript Chemerin in Volunteers and Patients with Insulin Resistance 2

and Diabetes” was aimed to investigate the level of chemerin in T2D pts and IR individuals. The authors included 18 participants with T2D and 116 without T2D. Insulin resistance was

measured by steady-state plasma glucose (SSPG) concentration during the insulin suppression test, which in time consuming test in clinical settings. The authors concluded that pts with T2D and those without T2D who were IR had the most proteolytic processing of chemerin resulting in higher levels of both cleaved and degraded chemerin. This suggests that increased inflammation in individuals who have T2D or are IR causes more chemerin processing.  

Comments to the Authors:

1.                  Title: please, redefine it, the expression volunteers is problematic, healthy subjects?

2.                  Abstract line 12: please, first sentence is not clear, be more specific abou the role of chemerin

3.                  Please define the aim of the study in Abstract section

4.                  Please put the study design in the Abstract section

5.                  In Methodology section please add permission to the authors that they can use data from Stanford University study about the role of IR in human diseases.

6.                  Please, add briefly in Methods study design and inclusion criteria for the study

7.                  Please add description of pts with T2D included in the study, especially about background antihyperglycemic medication they had

Author Response

Reviewer 2

This manuscript „Chemerin in Volunteers and Patients with Insulin Resistance 2

and Diabetes” was aimed to investigate the level of chemerin in T2D pts and IR individuals. The authors included 18 participants with T2D and 116 without T2D. Insulin resistance was

measured by steady-state plasma glucose (SSPG) concentration during the insulin suppression test, which in time consuming test in clinical settings. The authors concluded that pts with T2D and those without T2D who were IR had the most proteolytic processing of chemerin resulting in higher levels of both cleaved and degraded chemerin. This suggests that increased inflammation in individuals who have T2D or are IR causes more chemerin processing.

We appreciate the helpful comments from the reviewer and have answered the points below.

Comments to the Authors:

  1. Title: please, redefine it, the expression volunteers is problematic, healthy subjects?

The title has been changed to:” Chemerin in Participants with or without Insulin Resistance and Diabetes”

  1. Abstract line 12: please, first sentence is not clear, be more specific abou the role of chemerin

This sentence has been revised to read:“ Chemerin is a chemokine/adipokine regulating inflammation, adipogenesis and energy metabolism whose activity depends on successive proteolytic cleavages at its C-terminus.“

  1. Please define the aim of the study in Abstract section
  2. Please put the study design in the Abstract section

To answer both points 3 and 4, the sentence was revised to incorporate both the aim and the methods in the abstract:” This hypothesis was tested by characterizing different chemerin forms by specific ELISA in plasma of 18 participants with T2D and 116 without T2D who also had their insulin resistance measured by steady-state plasma glucose (SSPG) concentration during an insulin suppression test. This approach enabled us to analyze the association of chemerin levels with a direct measure of insulin resistance (SSPG concentration).”

  1. In Methodology section please add permission to the authors that they can use data from Stanford University study about the role of IR in human diseases.

The following has been added to the Methods section last sentence of paragraph 2.1:” …, and all individuals gave written informed consent to participate in the studies and for the use of their data in analyses regarding the role of insulin resistance in human disease.”

  1. Please, add briefly in Methods study design and inclusion criteria for the study

This information has been added to Section 2.1 Study Participants: “The study participants were in good general health and 18 to 80 years old. The participants with diabetes were either receiving treatment with one or more glucose lowering medications for management of hyperglycemia or had fasting glucose concentration ≥126 mg/dL on more than one occasion”

  1. Please add description of pts with T2D included in the study, especially about background antihyperglycemic medication they had.

Information about medications for participants with T2D has been added to Section 3.2 Levels of different chemerin forms in plasma: ”Of the 18 participants with DM2, eight were not receiving treatment with a glucose-lowering medication. Of the remaining 10 participants, seven were being treated with a sulfonylurea, two with metformin, and one with a thiazolidinedione.”

Reviewer 3 Report

Comments and Suggestions for Authors

Dear Authors,

This is interesting paper concerned problem of diagnosing insulin resistance. But I have a few suggestions /questions:

1.      In the introduction  should be explain the different forms of chemerin (not only 163S) and its biological differences.

2.      In the introduction should be explain the difficulties with insulin resistance diagnosis and then justify carrying out the study

3.      Material and methods: participants- this is the part where should be number of participants, inclusion and exclusion criteria for this study. Rules how participants were divided to the groups should be presented in this part.

4.      Material and methods participants – Mentioned one or more medication is too small information. This is completely different patient with type 2 diabetes , but lean, and treated with maximal doses SU, in comparison with obese patient with metformin, pioglitazone or GLP-1 agonist.- metabolically these patients are not comparable.

5.      Diabetes duration and status of metabolic control should be mention too.

6.      Material and methods: lines 91-93- this sentence is not logical- should be : diabetes diagnosing according ADA (…….), and treated  using ……, - diabetes group is so small that should be very strictly chosen and homogenous- if not this is serious limitation of the study

7.      Statistical analysis- in the paper Authors used parametrical methods of analysis- there is no information about normality of distribution and homogeneity of variance, and about testing it.

8.       Figure 6- comparing groups with and without diabetes- I think it should be better dividing on IS and T2D, and drawn separated curves for two groups- it would be better shown differences for readers

Author Response

Reviewer 3

Dear Authors,

This is interesting paper concerned problem of diagnosing insulin resistance. But I have a few suggestions /questions:

Thank you for the careful reading of our manuscript and the points raised which we have answered below.

  1. In the introduction should be explain the different forms of chemerin (not only 163S) and its biological differences.

The description of the different chemerin forms in the 2nd paragraph on page has been expanded to describe all of the different forms in our paper.

  1. In the introduction should be explain the difficulties with insulin resistance diagnosis and then justify carrying out the study

The aim of the study was to demonstrate if the claimed relationship between increased levels of chemerin and diabetes could be substantiated. The potential of using chemerin assays as a diagnostic tool for insulin resistance arose out of the analysis of the data. A new penultimate paragraph has been added to the introduction describing issues with the diagnosis of insulin resistance.

  1. Material and methods: participants- this is the part where should be number of participants, inclusion and exclusion criteria for this study. Rules how participants were divided to the groups should be presented in this part.

The second paragraph of section 2.2 describes the four groups with the added last sentence summarizing them.

  1. Material and methods participants – Mentioned one or more medication is too small information. This is completely different patient with type 2 diabetes , but lean, and treated with maximal doses SU, in comparison with obese patient with metformin, pioglitazone or GLP-1 agonist.- metabolically these patients are not comparable.

Of the 18 participants with DM2, eight were not receiving treatment with a glucose-lowering medication. Of the remaining 10 participants, seven were being treated with a sulfonylurea, two with metformin, and one with a thiazolidinedione. This information has been added to Section 3.2 Levels of different chemerin forms in plasma

  1. Diabetes duration and status of metabolic control should be mention too.

Unfortunately  information on duration since diagnosis before enrolling in the study was not available for the participants in the study. We have added this sentence to Section 2.1 to describe the status of metabolic control:”Participants with T2D had FPG levels of 179 ± 27 mg/dL (mean ± SD; range: 150- 258 mg/dL).

  1. Material and methods: lines 91-93- this sentence is not logical- should be : diabetes diagnosing according ADA (…….), and treated using ……, - diabetes group is so small that should be very strictly chosen and homogenous- if not this is serious limitation of the study

This paragraph has been clarified: “Participants with type 2 diabetes (T2D) composed the T2D group. They were those participants with a diagnosis of diabetes according to ADA criteria who were receiving treatment with one or more glucose lowering medications for management of hyperglycemia and/or had fasting glucose level ≥126 mg/dL on more than one occasion.”

  1. Statistical analysis- in the paper Authors used parametrical methods of analysis- there is no information about normality of distribution and homogeneity of variance, and about testing it.

The reviewer has raised an important point on the use of parametric analysis. To answer this point the paragraph on statistical methods (2.5) in the Methods section has now been expanded and the issues with non-constant variance stated.

  1. Figure 6- comparing groups with and without diabetes- I think it should be better dividing on IS and T2D, and drawn separated curves for two groups- it would be better shown differences for readers.

In figure 6, the participants with T2D are marked separately from the remaining participants but the regression analysis was performed with all of the data because we feel that gives a better representation of the effects of the full range of SSPG and BMI on chemerin and chemerin cleavage levels. The data for comparison of the IS group with T2D is in Figures 4 and 5 plus table 3.

Round 2

Reviewer 1 Report

Comments and Suggestions for Authors

Zhao et al showed two types of figures/tables in the revised manuscript. It is very difficult to read the manuscript in the form. As a conclusion, the relationship between insulin resistance and chemerin derivatives are not simple. Cleaved and degraded chemerin is increased in insulin sensitive subjects as well as insulin resistant subjects, indicating that chemerin derivatives are not good markers for insulin resistance. This reviewer also pointed that FPG is not an estimate of insulin resistance, but the authors did not analysis the data according to this point. The authors examined the roles of chemerin on insulin resistance in the present study, but the biological importance of chemerin processing is still largely unclear.  

Comments on the Quality of English Language

The English should be improved.

Reviewer 2 Report

Comments and Suggestions for Authors

The authors corrected the manuscript according to the suggestions. 

Author Response

Thanks for the comments.